# Tribological Properties of TiN Coating on Cotton Picker Spindle

Peng Pan [1,2], Jie Gao [1,2], Chaorun Si [3], Qiang Yao [1,2], Zhanhong Guo [1,2] and Youqiang Zhang [1,2,*]

1 School of Mechanical and Electrical Engineering, Tarim University, Alar 843300, China; 10757202201@stumail.taru.edu.cn (P.P.)
2 Modern Agricultural Engineering Key Laboratory at Universities of Education Department of Xinjiang Uygur Autonomous Region, Alar 843300, China
3 School of Mechanical Engineering, Northwestern Polytechnical University, Xi'an 710072, China
* Correspondence: zhangyq@taru.edu.cn

**Abstract:** The spindle is the key working part of the horizontal cotton picker, and the wear resistance of its surface directly affects the service life of the spindle. Improving the surface performance of the spindle is fundamental for improving the performance of cotton pickers. To enhance the wear resistance of the spindle surface, this study used the physical vapor deposition (PVD) technique to prepare TiN coating on the spindle substrate surface of the cotton-picking machine to improve the spindle surface rather than the original electroplated chromium coating. The microscopic morphology of the spindle was analyzed by scanning electron microscope (SEM), the mechanical and frictional properties of the spindle were tested by a nanoindentation tester and a friction wear tester, and the morphology of the worn spindle was observed by a portable microscope and a 3D surface profiler. The test results indicated that after the PVD treatment, the surface hardness of the spindle was about 2.5 times that of the electroplated chromium spindle, and the H/E value was 2.2 times that of the electroplated chromium spindle. PVD-TiN spindle showed better mechanical properties. In the friction test, under the same conditions, the wear rate of the PVD-TiN spindle was less than that of the chrome plating spindle. In a field test of 100 hm$^2$, the average wear area of the second tooth tip surface of the electroplated chromium spindle was about 2.17 times that of the PVD-TiN spindle. It was verified that the PVD-TiN spindle surface had better wear resistance than the electroplated chromium spindle. This study has certain research significance for the performance optimization of cotton pickers. Also, it is indicated that PVD-TiN coating can effectively improve the wear resistance of the spindle surface and provides a new method for enhancing the service life of the spindle.

**Keywords:** cotton picker spindle; physical vapor deposition; PVD-TiN coating; wear resistance; hardness





## 1. Introduction

With the rapid development of the cotton industry and machinery industry, China has become a major cotton-producing country, and Xinjiang, as China's main cotton-growing area, develop particularly rapidly. In 2022, Xinjiang's cotton planting area was about 2.497 million hm$^2$, with a total output of 5.391 million tons, accounting for 90.2% of the total national cotton output and more than 20% of the global cotton output [1]. To reduce human and material resources and save picking costs and improve picking efficiency, mechanical picking is the main way of cotton picking. There are more than 7000 cotton pickers in Xinjiang, and the machine picking rate is more than 80% [1]. As a key component of the cotton picker, the spindle has a short service life (about 400 hm$^2$ under normal conditions) and a large number of replacements (more than 3000 spindles are installed on a six-row cotton picker), and it can only be used as scrap after replacement, which cannot be recycled, resulting in extremely serious economic losses. Meanwhile, the wear and tear of the spindle also lead to a gradual decline in the harvesting efficiency and seed quality of the cotton during the picking process. To improve picking efficiency, extend the service

life of the spindle and reduce the cost of cotton harvesting, it is necessary to improve the strength and wear resistance of spindles.

The general method to improve the wear resistance of metal surfaces is to prepare wear-resistant coating on the surface to enhance hardness [2]. Electroplated chromium coating has been used as a wear-resistant layer on the surface of spindles because of its low production cost, convenient operation, and high performance. However, it will cause serious environmental pollution and affect human health in the production process, and there are huge dense micro-cracks on the surface, which will have serious extension and propagation behaviors during friction and eventually cause the coating to crack or fall off [3,4]. In recent years, to improve spindle performance, studies are mainly conducted on spindle wear failure [5–7], simulation analysis [8,9], system design [10,11], and other aspects, but little research has been conducted on coating modification of spindle surface. Meng [12] investigated the friction properties of nickel plating and chromium plating coating on the spindle surface. The results indicated that the friction coefficient of nickel plating coating is lower than that of chromium plating coating, and it gradually decreases with the increase of rotating speed and load, but the wear resistance of chromium plating coating is better than that of nickel plating coating. Zhang [13] adopted the electromagnetic strengthening technology to treat the spindle surface and compared its performance with that of the untreated spindle. The results showed that the width of the grinding mark of the spindle after treatment is significantly smaller than that of the untreated chrome spindle, and the residual stress is greatly reduced, indicating that the electromagnetic treatment applied to the spindle surface can improve the anti-wear ability of the spindle surface. Amanov [14] exploited ultrasonic nanocrystal surface modification technology to treat a spindle and compared it with an untreated spindle through friction and wear tester. The results demonstrated that the surface roughness of the spindle after treatment decreases, and the hardness and wear resistance increase, indicating that this method can enhance the service life of the spindle. Although these studies have improved the performance of the spindle, successful replacement of chromium plating coating on the spindle surface has not been reported.

In the context of advocating environmental protection globally, chrome replacement technology is always a scientific research hotspot and has become more mature [15]. At present, PVD technology is one of the most effective technologies for surface modification in surface engineering, and it is also one of the most promising alternative processes for chromium plating [16,17]. The coatings prepared by PVD technology have characteristics of high hardness, good wear resistance, low friction coefficient, dense tissue structure, and strong bonding with the substrate [18,19]. Besides, the prepared coatings have low surface roughness and generally do not require secondary polishing treatment, providing an ideal solution for small shaped parts that are difficult to polish. In recent years, there are more and more studies and achievements of PVD technology instead of chromium, and numerous tests have proved that PVD coating has excellent performance and can completely replace the electroplating process. By using the PVD process, Picas [20] deposited nitrides on 6063 nickel-plated aluminum substrates and compared the mechanical and tribological properties with those of electrochromic coatings, which proved that the hardness, adhesion, and wear resistance of PVD ceramic coatings are superior to those of electrochromic coatings. Daure [16] deposited chromium coatings on the surface of 316 stainless steel by employing PVD and electroplating techniques, and the results indicated that the coatings prepared by the PVD technique could replace electroplated chromium layers in some tribological applications.

As a representative of hard coating, TiN has good strength, hardness, oxidation resistance, and wear resistance, so it is widely used in the mechanical industry [21–23]. The PVD-TiN coating has a very high hardness. According to the wear amount equation given by Archard [24], the material surface hardness is inversely proportional to the wear amount. From this perspective, PVD-TiN coating can effectively reduce the amount of material wear and significantly enhance the wear resistance and service life of the material.

Surface coating modification tests using PVD technology instead of electrochromism plating technology can solve the problem of serious environmental pollution caused by the chromium production process and avoid the negative effects of micro-cracking caused by electrochromism plating coatings due to hydrogen embrittlement [25]. This study presents a new solution for improving the surface strength of spindles, has some research significance to improve the wear resistance and service life of spindles, and provides a reference for the research on PVD-TiN coating instead of electroplated chromium coating series, which is consistent with the development direction of green manufacturing.

## 2. Wear Mechanism of Spindle and Coating Preparation

### 2.1. Structure of Spindle

The test substrate and chromium plating spindle are produced by a reliable company, Reliable Tech in Chengdu, China. The matrix of the spindle is shown in Figure 1. The working part of the spindle picking is conical, the middle support part is cylindrical, and the tail end is driven by bevel gear meshing. The total length of the spindle is about 121 mm, and the diameter of the middle rod part is about 12.3 mm (approximately 12.38 mm after chrome plating). To improve the ability to pick cotton in the field, the production and processing of the spindle need to form hook teeth with an inclination angle on the conical surface through mechanical cutting, so the base material of the spindle is chosen to be low-carbon alloy steel with good machinability. The base material for spindles in the current market is generally 20CrMnTi, which is characterized by good machinability and fatigue resistance, high hardenability and impact toughness, and is suitable for manufacturing high-strength parts of medium and small sizes [26]. The chemical elemental composition of 20CrMnTi are shown in Table 1.

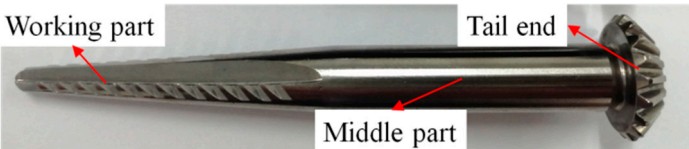

**Figure 1.** The matrix of the spindle.

**Table 1.** The chemical elemental composition of 20CrMnTi (wt.%).

| Ciomposition | C | Si | Mn | Cr | Ni | Cu | Ti | S | P |
|---|---|---|---|---|---|---|---|---|---|
| wt.% | 0.196 | 0.229 | 0.954 | 1.192 | 0.031 | 0.028 | 0.048 | 0.0056 | 0.014 |

### 2.2. Wear Mechanism of Spindle

During the use of the workpiece, the damage and failure usually start from the surface [20]. In the cotton-picking process of the cotton picker, due to the long-term contact and friction between the spindle and the cotton and the cotton rod, the spindle will gradually wear out and fail, which is mainly reflected by the reduction of the roughness. Meanwhile, in the process of cotton stripping, the spindle is subject to the extrusion friction of the cotton stripping disc, which is much larger than the picking friction, and this is also an important reason for the wear failure of the spindle. Additionally, Xinjiang has a bad ecological environment. There are often many dust and sand particles on the cotton. These impurities with higher hardness will cause more serious abrasive wear to the spindle in the process of picking and winding the cotton fiber, which will directly lead to the spindle breaking in serious cases [27,28]. Moreover, in terms of the friction and wear degree of the spindle, the head is more serious than the tail, and the tooth surface is more serious than the conical surface, which is caused by the fact that the head of the spindle keeps picking and its linear speed is greater than that of the tail. When the wear-resistant layer on the spindle surface is torn to expose the substrate material, the substrate will undergo oxidation wear, and the oxide particles will gradually form and peel off during the picking process, causing the expansion of the spindle surface to form a "broom type" wear pattern [6].

### 2.3. The PVD Technique

The coating is prepared by using multi-arc ion plating equipment (IKS, Shenyang, China), and the target material is titanium metal target with a purity of 99.99%. The main process parameters are set as follows: the current is 80 A, the voltage is 20 V, the substrate bias is –100 V, and the duty cycle is 60%. Before placing the substrate in the vacuum chamber, the clean surface is obtained by ultrasonic cleaning in anhydrous ethanol. After drying, to further clean the substrate surface, it is bombarded by electromagnetic fields in the vacuum chamber to remove dirt and impurities. The deposition temperature is set to 200 °C, and a lower deposition temperature has a smaller impact on the matrix performance. The reaction gas introduced is $N_2$. The test sample is shown in Figure 2.

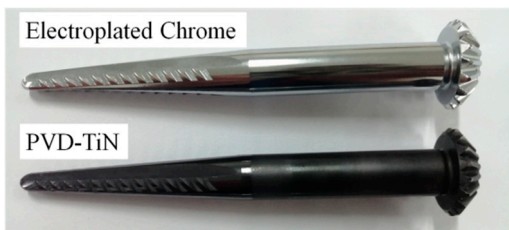

**Figure 2.** A photograph of the spindles with two coating technique.

The SuperView 3D surface profiler (CHOTEST, Shenzhen, China) is employed to measure the surface roughness of ten randomly selected spindles (five for each of the two types of spindles). Each sample is tested at a random position seven times, and the average value is taken after removing the extreme value. Through the test and analysis, the surface roughness Ra of the PVD-TiN spindle and the chrome plating spindle is 0.381 and 0.503 μm, respectively. The test morphology is shown in Figure 3.

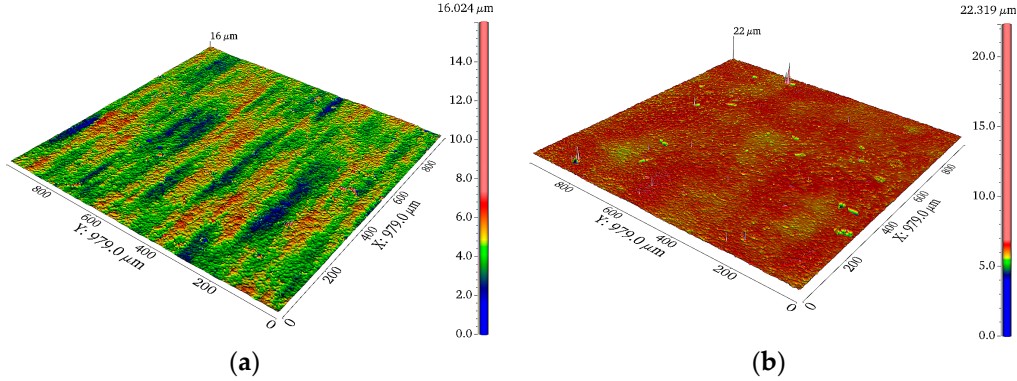

(**a**)                        (**b**)

**Figure 3.** The 3D surface micromorphology for the two tested surfaces (**a**) the PVD-TiN spindle (**b**) the electroplated chrome spindle.

## 3. Materials and Methods

### 3.1. Specimen Preparation

The PVD-TiN and chromium plating spindles were selected as samples. Several sets of specimens were cut out from the samples by using METCUT-8 metallographic cutting machine, and some of them were selected for microscopic morphology and friction tests respectively. Due to the irregular size of the sample during section observation, the observation effect is greatly affected, so the observation sample was pretreated. First, the sample was cold-inlaid with metallographic resin and then polished with a metallographic abrasive paper with a mesh size from coarse to fine to 2000 mesh. After the test sample with a smooth surface was obtained, the average particle size was 3 and 1 μm diamond suspension for preliminary polishing, and then the average particle size of 0.04 μm of $SiO_2$ suspension was used for further polishing.

### 3.2. Microstructure Test

The microstructure of two types of spindle specimens was photographed several times by SEM (TM4000, Hitachi, Tokyo, Japan), the surface morphology structure of both coatings was compared and analyzed, and the coating composition was examined by Energy Dispersive Spectroscopy (EDS) (JEOL, Tokyo, Japan). Finally, the cross-sectional morphology, thickness, and bonding of the coatings were observed.

### 3.3. Mechanical Property Test

The microhardness and the elastic modulus of the sample surface were tested by a high-precision nanoindentation tester (TI980, Bruker, Billerica, MA, USA), the test load was 10 mN, and the holding time was 2 s. To reduce the error, nine samples were taken from each of the two main shaft samples, the single sample was tested three times, and the mean value was taken.

### 3.4. Tribological Test

During the use of the spindle, the wear situation will increase with time. In this study, dry friction tests were conducted on several groups of PVD-TiN and chromium-plated spindle samples by using MXW-1 friction and wear tester (YIHUA, Jinan, China). To reduce the influence of surface roughness on the test results, the surface roughness of the chromium-plated spindle should be polished to 0.38 μm before the friction test, which was close to that of the PVD-TiN spindle. The samples were ultrasonically cleaned in anhydrous ethanol for 10 min before and after the test to remove any residual contaminants, and then they were dried in air. The working mode was selected as reciprocating friction motion, the test normal load was 5 N, the frequency was 5 Hz, the displacement was 2000 μm, and the upper specimen substrate was selected of $Si_3N_4$ grinding balls with a diameter of 6.35 mm and a hardness of approximately 78 HRC. The friction time was set to 30, 60, and 90 min, respectively. A portable microscope was used to observe the surface abrasion shape of the specimen at a magnification of 200×. The wear scar pattern of the sample was observed at 10× under the objective lens using a 3D surface profiler to measure the amount of wear. The wear rate W was calculated according to Equation (1).

$$W = V/(F \times d) \tag{1}$$

where W is the specimen wear rate in $mm^3/(N \cdot m)$, F is the normal load applied in the test in N, d is the travel distance in m, and V is the loss volume in $mm^3$.

During the cotton-picking process, the spindle is in contact with the debris of different hardness causing different contact forces, and the picking efficiency is greatly affected by the spindle speed. Considering that a lower friction coefficient can effectively reduce the frictional power consumption during operation [29], to investigate the loss of frictional power consumption of the spindle by contact force and rotational speed, friction tests were conducted on two types of spindle specimens under different loads and frequencies. The test results were only analyzed for the friction coefficient. The friction time was 10 min, and the displacement was 1000 μm. Other test preparations and conditions remained unchanged. The working principle of the friction test is shown in Figure 4.

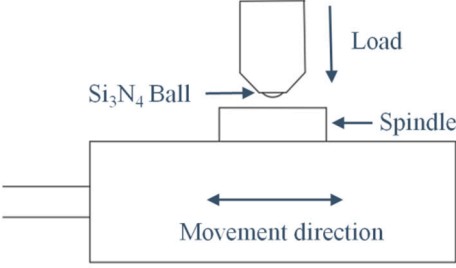

**Figure 4.** The working principle of the friction test.

## 4. Results and Discussion

### 4.1. Microstructure Analysis

The coating morphology can largely reflect the coating properties, and generally, the denser the coating surface morphology and the fewer cracked pores, the better the coating mechanical properties [30]. As for the surface topography of the PVD-TiN spindle, it can be seen that the coating has a compact structure, uniform distribution of surface particles, "island" characteristics, and no obvious pores and cracks. In contrast, there are obvious defects such as microcracks and concave holes in the chrome plating spindle, and microcracks are the main reason that affects the surface performance of the chrome plating spindle [3,20]. The PVD-TiN spindle coating cross-section showed no obvious defects, the deposited layer was well bonded to the substrate, and a single-layer coating with a thickness of about 11 µm and a continuous structure was prepared. Additionally, the coating thickness of the electroplated chromium spindle was about 38 µm. The surface and cross-sectional morphology of the two samples are shown in Figures 5 and 6, respectively.

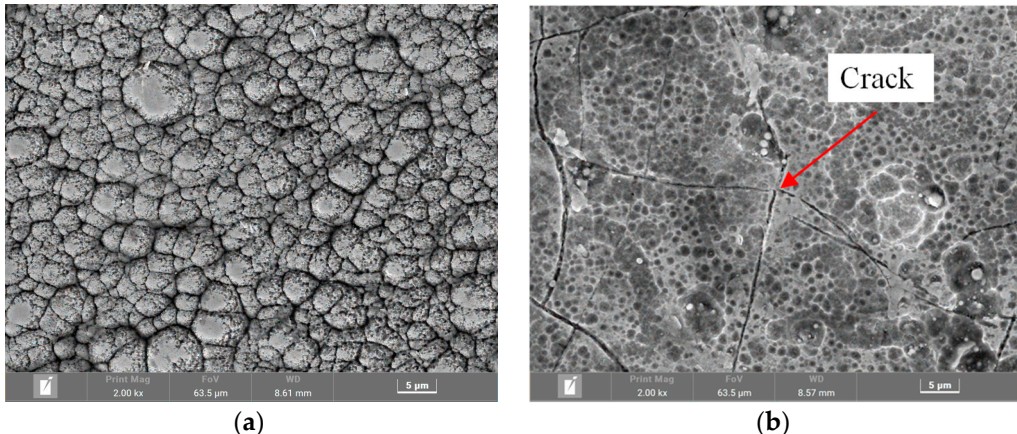

**Figure 5.** The SEM images for the two tested surfaces (**a**) the PVD-TiN spindle (**b**) the electroplated chrome spindle.

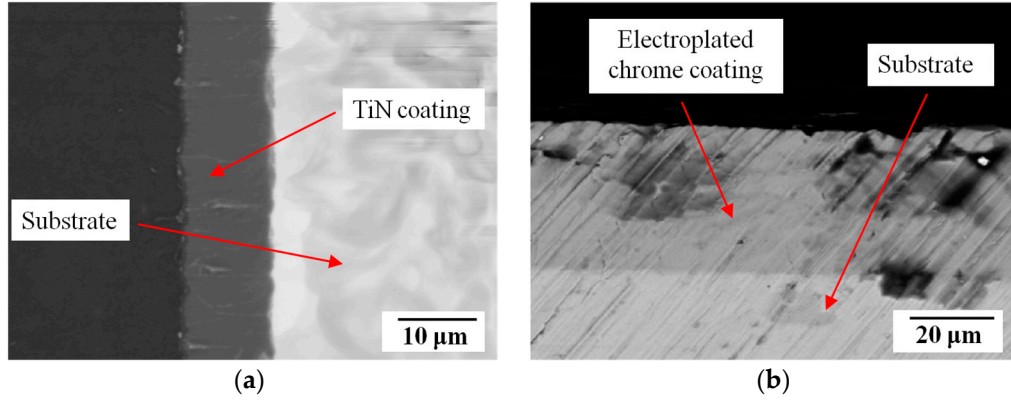

**Figure 6.** The section morphology of SEM for the two tested surfaces (**a**) the PVD-TiN spindle (**b**) the electroplated chrome spindle.

The elemental composition of the two coatings was analyzed by EDS (as shown in Table 2). The elemental mass ratios of Ti and N in the TiN coating were 78.17% and 21.83% respectively, which showed that the atomic number was close to 1:1, proving that the main component was TiN; meanwhile, the elemental mass percentage of Cr in the electroplated chromium coating was 96.25%, indicating that the main component was Cr and the rest is C.

**Table 2.** The chemical composition of the spindle in wt.%.

| Elements | Ti | N | Cr | C | Total |
|---|---|---|---|---|---|
| PVD-TiN | 78.17 | 21.83 | - | - | 100.00 |
| Electroplated chrome | - | - | 96.35 | 3.65 | 100.00 |

### *4.2. Mechanical Performance Analysis*

Figures 7 and 8 show the elastic modulus and hardness of the two specimens. The modulus of elasticity and hardness are important indicators for evaluating the mechanical properties of coatings, and generally, the higher the hardness, the better the wear resistance of the material [20]. It can be seen from Table 3 that the mean value of the nano-hardness of the PVD-TiN spindle surface is 20.57 GPa, which is about 2.5 times that of the electroplated chromium spindle. The ratio of the modulus of elasticity to the hardness of a material (H/E) measures the relative wear resistance, and the larger the H/E value, the better the relative wear resistance and resistance to elastic strain [31]. The results in Table 3 indicate that the H/E value of the PVD-TiN spindle is 0.0937, which is about 2.2 times that of the electroplated chromium spindle, indicating better relative wear resistance.

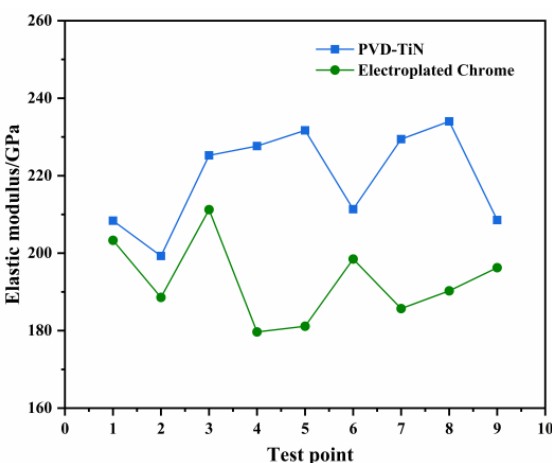

**Figure 7.** The elastic modulus for both case of coating layers.

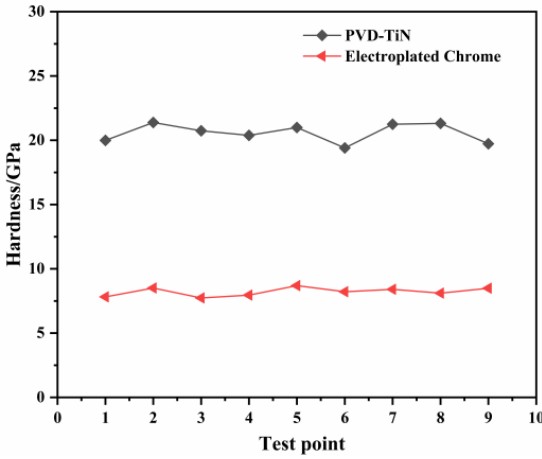

**Figure 8.** The hardness for both case of coating layers.

**Table 3.** The elastic modulus and hardness.

| Type | Value | Elastic Modulus/Gpa | Hardness/Gpa | H/E |
|---|---|---|---|---|
| Electroplated chrome | Max | 211.24 | 8.69 | - |
| | Min | 181.13 | 7.73 | - |
| | Mean | 192.73 | 8.21 | 0.0426 |
| | Standard deviation | 10.48 | 0.34 | - |
| PVD-TiN | Max | 234.03 | 21.38 | - |
| | Min | 199.27 | 19.41 | - |
| | Mean | 219.52 | 20.57 | 0.0937 |
| | Standard deviation | 12.63 | 0.73 | - |

The area enclosed by the loading and unloading curve and the displacement distance in Figure 9 represents the energy absorbed by the coating, i.e., the plastic deformation work. Higher plastic deformation work indicates that more severe plastic deformation is produced [32]. The plastic deformation work of the electroplated chromium spindle in the figure is 2.38 times that of the PVD-TiN spindle, showing that the plastic deformation of the electroplated chromium spindle occurs more.

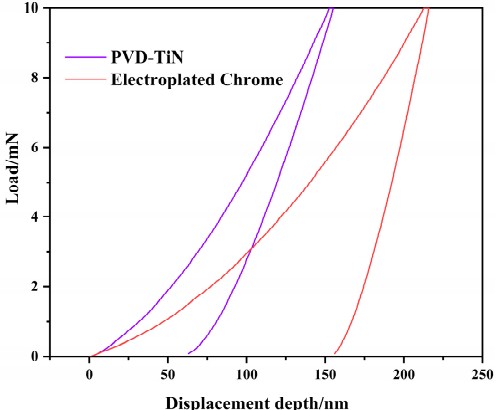

**Figure 9.** The load-depth curve for both case of coating layers.

### 4.3. Friction and Wear Behavior

### 4.3.1. Friction Coefficient Analysis

### At Different Times

As shown in Figure 10, the friction coefficient of the PVD-TiN spindle friction test gradually increased with the friction time before 2200 s. The main reason may be that the surface of the TiN coating is dense and its hardness is much higher than that of the $Si_3N_4$ grinding ball. As the friction time increased, the contact area between the $Si_3N_4$ ball and the sample gradually increased, and there were more contact roughness peaks, resulting in a gradual increase in the friction coefficient. It can be seen from the test stage of 2200–3500 s that the friction coefficient tended to be stable at about 0.64 with the increase in the friction time, and then it decreased slowly. This was due to the removal of the surface roughness peak of the object to be worn during the wear process. As the roughness peak was removed from the contact body, the surface roughness peak of the object to be worn became flat, leading to a lower friction coefficient. At the test stage of 3500–4250 s, the friction coefficient of the PVD-TiN spindle fluctuated greatly and reached the maximum value, which may be caused by the presence of hard TiN particles in the abrasive debris, which significantly increased the friction coefficient of three-body wear between the abrasive object and hard particles. At this time, the wear mechanism was mainly abrasive wear.

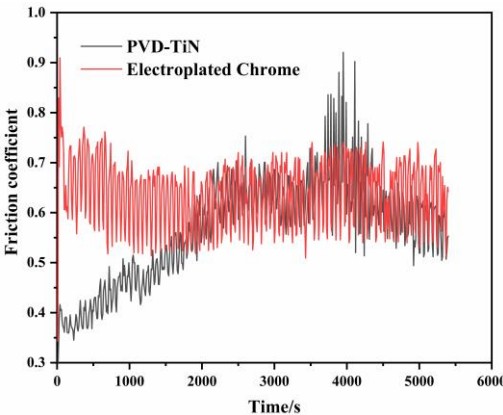

**Figure 10.** The change in the friction coefficient with time.

The friction coefficient of the chromium-plating spindle before 1000 s of the test decreased slowly with the increase in the friction time, which may be due to more defects on the surface of the chromium-plated spindle. After 1000 s of the test, the coefficient of friction leveled off and finally stabilized at about 0.62. The friction coefficient of the chromium-plated spindle fluctuated greatly and stabilized more rapidly, and this was probably because the hardness of $Si_3N_4$ on the grinding ball was greater than that of the chromium-plated coating, resulting in the presence of numerous Cr particles on the coating surface during friction; at this time, the wear mechanism was mainly adhesive wear.

At Different Loads

In this test, the friction frequency was set to 5 Hz, and the normal load was set to 10, 20, and 30 N, respectively.

During harvesting, since the spindle not only contacts with cotton but also with cotton stalks and sand in the air, this section investigates the effect of different contact loads on the spindle. Figure 11 shows that the friction coefficient of both types of spindles decreased with the increase in the contact load under the same friction time. This is because the increase in load caused an increase in the contact depth and contact area of the spindle surface and a decrease in unit pressure, resulting in a decrease in the coefficient of friction. The comparison between Figure 11a,b shows that the friction coefficient of the PVD-TiN spindle specimens under the same load and the same friction time within 10 min of the friction test was smaller than that of the electroplated chromium spindle, proving that the frictional power consumption of PVD-TiN spindle was smaller.

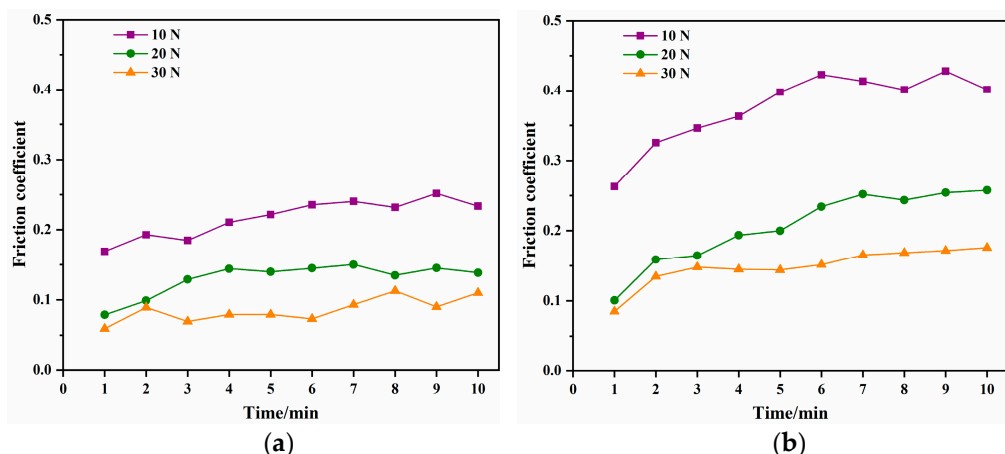

**Figure 11.** The friction coefficient changes under different loads (**a**) the PVD-TiN spindle (**b**) the electroplated chrome spindle.

At Different Frequencies

In this test, the normal load was set to 5 N, and the friction frequency was set to 3, 4, and 5 Hz, respectively.

Figure 12 shows that under the same friction time, the friction coefficient of the two types of spindles increased with the friction frequency, and this may be because as the friction frequency increased, the abrasive particles on the spindle surface were partially thrown out, reducing surface friction and scratching; as a result, the surface was smoother, so the surface friction coefficient was reduced. The comparison between Figure 12a,b shows that under the same friction frequency within 10 min of friction test, with the increase in the friction time, the friction coefficient of the PVD-TiN spindle was basically smaller than that of the electroplating chromium spindle, which proved that the frictional power consumption of PVD-TiN spindle was smaller. The reason why the friction coefficient of the PVD-TiN spindle was slightly higher than that of the electroplated chromium spindle at individual time points may be due to TiN abrasive chips, which are harder and lead to increased surface scratching and wear, resulting in an increased friction coefficient.

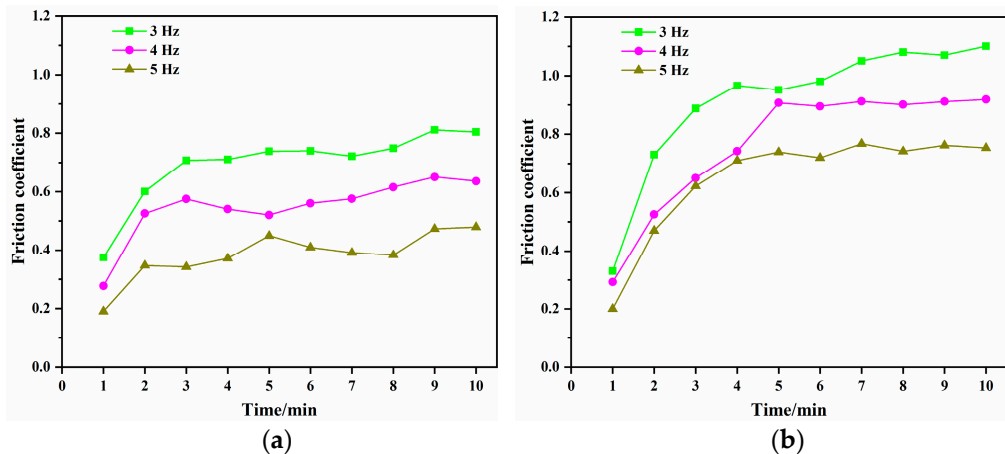

**Figure 12.** The friction coefficient changes under different frequencies (**a**) the PVD-TiN spindle (**b**) the electroplated chrome spindle.

### 4.3.2. Wear Morphology Analysis

Figure 13 shows the morphology of the grinding marks of the PVD-TiN and chromium plating spindles observed with a portable microscope at different friction times. In Figure 13, (1) denotes the PVD-TiN spindle, and (2) denotes the chrome plating spindle. It can be seen from the figure that the wear scar morphology of the PVD-TiN and chromium plating spindles gradually broadened with the increase in friction time. At the same friction time, the width of wear marks on the surface of the PVD-TiN spindle was smaller than that of the chrome plating spindle, and there was no obvious wear furrow. Besides, the surface of the chromium plating spindle was more seriously damaged, and there were many obvious grooves in the center of the wear mark.

Figure 14 shows the 3D wear trace topography of the PVD-TiN and chromium-plated spindles under different friction time by a 3D surface profiler. In Figure 14, (1) denotes the PVD-TiN spindle, and (2) denotes the chrome plating spindle. Though the color scale at the right column in the figure cannot represent the actual wear depth, it can be seen that with the increase in friction time, the depth of the grinding marks of the PVD-TiN and chromium plating spindles gradually deepened, and the damage gradually became serious.

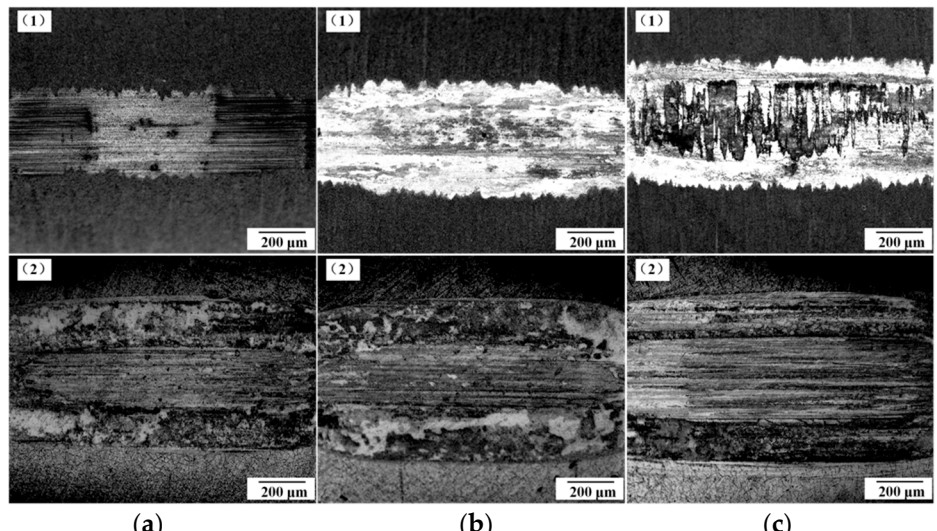

**Figure 13.** The SEM micrographs of the wear surfaces with different time. (**a**) Wear for 30 min (**b**) Wear for 60 min (**c**) Wear for 90 min.

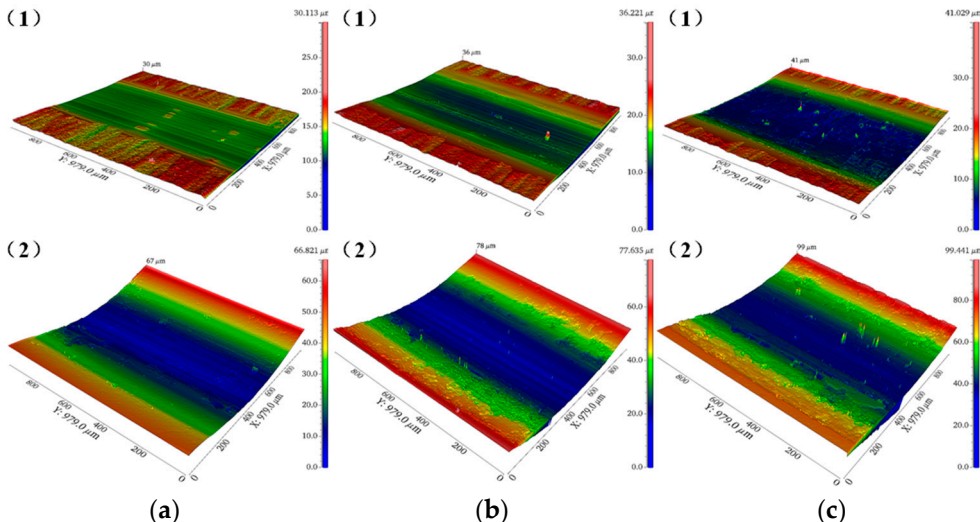

**Figure 14.** The wear trace topography of the wear surfaces with different time. (**a**) Wear for 30 min (**b**) Wear for 60 min (**c**) Wear for 90 min.

Figure 15 presents the change curve of the wear trace profile of the PVD-TiN and electroplated chromium spindles under different friction time. It can be seen that the grinding mark depth of both types of spindles increased with the friction time. At the same friction time, the profile curve of the wear mark of the electroplated chromium spindle was much lower than that of the PVD-TiN spindle. The maximum abrasion depth of the electroplated chromium spindle was about 48 μm at 90 min of friction, while that of the PVD-TiN spindle was less than 15 μm.

Figure 16 shows the EDS composition analysis of the center of the surface abrasion marks after 90 min of wear of the two types of spindle tests. As shown in the figure, both specimens had a large number of Fe elements, and the center of the surface abrasion marks leaked out of the matrix at this time, which was consistent with the results of the abrasion marks profile. The smaller figures on the top right is the test area.

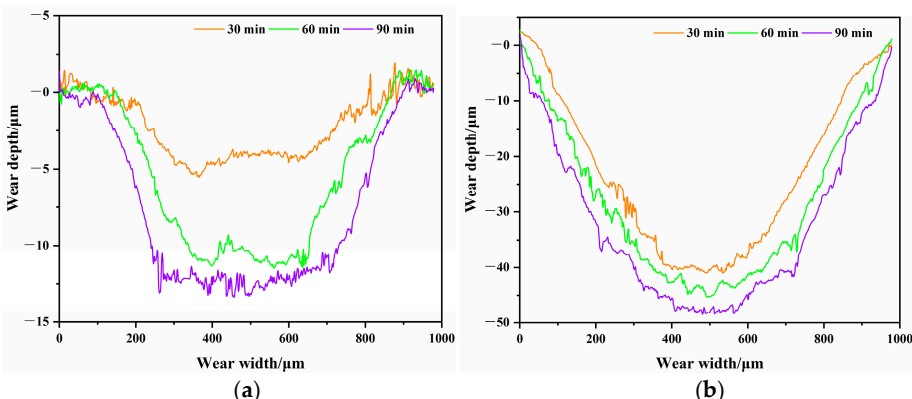

**Figure 15.** The wear profile of the wear surfaces with different time. (**a**) The PVD-TiN spindle (**b**) The electroplated chrome spindle.

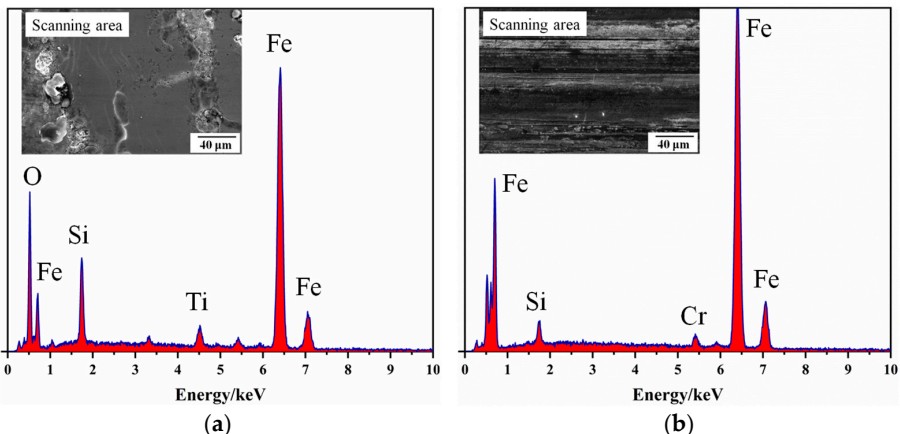

**Figure 16.** EDS analysis of the spindle after 90 min for the wear center (**a**) The PVD-TiN spindle (**b**) The electroplated chrome spindle.

### 4.3.3. Analysis of Wear Results

Figure 17 illustrates the wear rates of the PVD-TiN and electroplated chromium spindles at different friction time. It can be seen from the figure that the wear rate of the PVD-TiN spindle was much lower than that of the electroplating chromium spindle. For example, at the friction time of 90 min, the wear rate of the electroplating chromium spindle was about 5 times of that of the PVD-TiN spindle, 7.09 and 1.41 respectively. The test proved that the wear resistance of the PVD-TiN spindle surface is much better than that of the chromium-plating spindle, and the PVD-TiN coating greatly improves the wear resistance of the spindle surface.

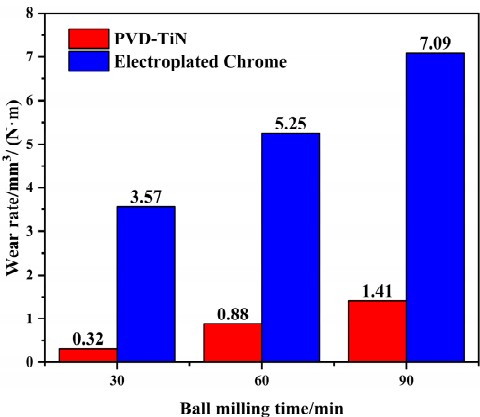

**Figure 17.** The wear rate of the spindle with different time.

*4.4. Field Test and Results*

During actual working, the spindle is always in contact with "soft" materials such as cotton. To determine the actual wear resistance of the PVD-TiN spindle, two types of spindles were tested and analyzed in field tests to compare and analyze the wear resistance performance.

Generally, the cotton picker drum in the upper position spindle has more contact with the cotton, and the lower part has more contact with the cotton pole. To increase the accuracy of the wear test after the picking work for both types of spindles, the PVD-TiN and electroplating chromium spindles were installed inside the same front drum in the upper middle position of the cotton picker before the field test. The test site is located at the 10th Regiment Farm of Alar City, the first division of Xinjiang Production and Construction Corps, China. The trial was conducted in mid to late October 2022, with a total trial area of 100 hm². The flow of the field test is shown in Figure 18, and the yellow arrow in the figure indicates the direction of the test flow sequence.

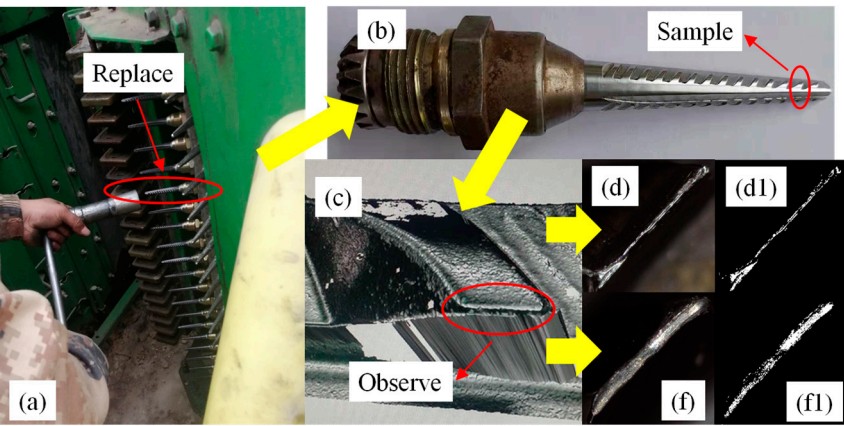

**Figure 18.** The flow chart of field test (**a**) The spindle sampling position (**b**) The hook tooth sampling position (**c**) The hook tooth shooting position (**d**) The PVD-TiN tooth tip morphology (**d1**) The PVD-TiN tooth tip morphology after LabVIEW processing (**f**) The chrome plating tooth tip morphology (**f1**) The chrome plating tooth tip morphology after LabVIEW processing.

Since the first three hook teeth of the spindle wear more seriously, and the first tooth wears more than the rest of the teeth by the rest of the debris contact force, serious individual wear or broken teeth is easy to occur; meanwhile, the wear of the third hook tooth wear is significantly smaller than the second hook tooth, so a portable microscope was exploited in this study to observe the second tooth at the same installation position of the two spindles for multiple comparative photographies. After shooting, the image was processed by LabVIEW, the hook tooth wear area was calculated by pixel points, and the final result was averaged. This method measured the wear resistance of two types of spindles by determining the area of the tip of the spindle teeth after wear. The calculation results indicated that the average wear area of the second hook tooth of the electroplated chromium spindle was about 2.17 times that of the PVD-TiN spindle, and the wear of the electroplated chromium spindle was more serious than that of the PVD-TiN spindle. The test further verified that the wear resistance of the PVD-TiN spindle was far better than that of the electroplated chromium spindle, which verified the feasibility of this study and provided theoretical guidance for the optimization of spindles.

During the field test, the PVD-TiN spindle did not break teeth or coating peeling. However, due to force majeure factors, long and large-scale harvesting trials were not conducted, and the data are not yet available for accurate analysis of the cotton-picking net rate, trash content rate, and the service life of the spindle. Subsequently, different regions and cotton varieties will be selected for large-scale practical validation. During the test, some PVD-TiN spindles shook violently, which was caused by the poor fit between the spindle and the sleeve, and the subsequent assembly needs to be tested.

## 5. Conclusions

1.  The PVD-TiN spindle has a tight surface structure, with uniform particle distribution and "island" characteristics and no obvious pores and cracks. There are obvious defects such as microcracks and pits in the chrome plating spindle. The surface roughness Ra of the PVD-TiN and electroplating chromium spindles is 0.381 and 0.503 μm, respectively.
2.  The surface nano-hardness of the PVD-TiN spindle is 20.57 GPa, which is about 2.5 times that of the electroplated chromium spindle, and the H/E value is 0.0937, which is about 2.2 times that of the electroplated chromium spindle. These results indicate that the PVD-TiN spindle has better surface mechanical properties and better resistance to plastic deformation.
3.  In the friction test, the friction coefficient of the PVD-TiN spindle under the same conditions was smaller than that of the electroplated chromium spindle, which proved that the PVD-TiN spindle had lower frictional power consumption. The wear mechanism of the PVD-TiN spindle was mainly abrasive wear, and the wear mechanism of the chromium-plating spindle was mainly adhesive wear. With the increase in friction time, the wear scar shape of the PVD-TiN and chromium plating spindles gradually deepened and widened. However, the abrasion depth on the surface of the PVD-TiN spindle was much smaller than those of the chromium-plating spindle, and no obvious furrow appeared on the wear surface. When the friction was 90 min, the PVD-TiN spindle wear rate was about 1/5 of the electroplating chromium spindle. The wear resistance of the PVD-TiN spindle was far better than that of the chrome plating spindle.
4.  In the field test with an area of 100 hm$^2$, the average wear area of the second hook tooth of the electroplated chromium spindle was 2.17 times that of the PVD-TiN spindle. The test results indicate that the PVD-TiN spindle has better wear resistance than the chrome-plating spindle.

**Author Contributions:** Conceptualization, Y.Z. and C.S.; methodology, Y.Z.; software, Z.G. and P.P.; validation, Y.Z., C.S. and P.P.; formal analysis, Y.Z.; investigation, P.P.; resources, Y.Z.; data curation, P.P. and Q.Y.; writing—original draft preparation, P.P.; writing—review and editing, Y.Z., J.G. and P.P.; supervision, Y.Z.; project administration, Y.Z.; funding acquisition, Y.Z. All authors have read and agreed to the published version of the manuscript.

**Funding:** This work was financially supported by the National Natural Science Foundation of China (Grant No. 12262034), Bintuan Science and Technology Program (Grant No. 2021CB036), Youth Programs of Tarim University President Funded (Grant No. TDZKSS202111).

**Institutional Review Board Statement:** Not applicable.

**Informed Consent Statement:** Not applicable.

**Data Availability Statement:** The data presented in this study are available on-demand from the first author at (10757202201@stumail.taru.edu.cn).

**Acknowledgments:** The authors would like to thank their schools and colleges, as well as the funding of the project. All support and assistance are sincerely appreciated. Additionally, we sincerely appreciate the work of the editor and the reviewers of the present paper.

**Conflicts of Interest:** The authors declare no conflict of interest.

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
