# Peer review of "Tribological Properties of TiN Coating on Cotton Picker Spindle"

_coatings, doi:10.3390/coatings13050959_

Round 1
Reviewer 1 Report
Tribological properties of TiN coating on cotton picker spindle
Some notes:
1. Line 143: It is important to mention how to prepare the surface of the spindle before the coating process.
2. Lin 153: Figure 2. The image of the spindle → Figure 2. A photograph of the spindles with two coating technique.
3. Line 160: metallographic cutting machine, which type of cutting machine did you use? Describe it.
4. Line 163: 2.2 Micromorphology test → 2.2 Microstructure test
5. Line 179: the results were averaged → the mean value was taken, In this way, a
total of nine groups of data were obtained. Delete it
6. Should be ground to 0.38 μm → should be polished to 0.38 μm
7. Line 194; magnification of 200. → magnification of 200×.
8. Line 227: Figure 5. The surface morphology of SEM → Figure 5. The SEM images for the two tested surfaces
9. Lines 229-230: The elemental 229 mass percentages of Ti and N in the TiN coating were 78.17 and 21.83 respectively. Indicate the source of nitrogen in the coating layer of TiN. This is a very significant point.
10. Line 233: indicating that the main component was Cr.
→ indicating that the main component was Cr and the rest is C.
11. Line 238: that the average nano-hard → that the mean value of the nano-hardness
12. Line: 247: Figure 7. The elastic modulus → Figure 7. The elastic modulus for both case of coating layers.
Please reconsider all Figures’ captions for better and clearer description.
13. Linse 301-302: Figure 11 . Please make the curves sharper and clearer, and same for Fig. 12, Fig. 15, and Fig 11.
14. Line 335: Figure 13. The surface wear morphology → The SEM micrographs of the wear surfaces with different time.
15. Line 368: TiN spindle was far lower → TiN spindle was much lower
Grammar check.
Structure check.
Use cmprehensive and clear captions for figures and tables.
Author Response
Dear reviewer:
Thank you for your comments concerning our manuscript entitled “Tribological properties of TiN coating on cotton picker spindle” (coatings-2356466). Those comments are valuable and very helpful for revising and improving our manuscript. We have studied the comments carefully and have made the correction. The responses to the comments are as follows:
Point 1: Thank you very much for your constructive comments, we apologize for not introducing this point and have supplemented the article accordingly. The following content has been supplemented to the manuscript: Before placing the substrate in the vacuum chamber, the clean surface is obtained by ultrasonic cleaning in anhydrous ethanol. After drying, to further clean the substrate surface, it is bombarded by electromagnetic fields in the vacuum chamber to remove dirt and impurities.
Response 1: It is important to mention how to prepare the surface of the spindle before the coating process.
Point 2: Firstly, we apologize deeply for the confusion caused by our lame presentation ability. Thank you very much for your modification, which will greatly increase the readability of the manuscript.
Response 2: Figure 2. The image of the spindle → Figure 2. A photograph of the spindles with two coating technique.
Point 3: We apologize for not providing a description of the instruments used. All instrument names have been added to the manuscript. Thank you for your constructive comments. The cutting machine used is the MetLab low-speed precision cutting machine METCUT-8 from the United States.
Response 3: metallographic cutting machine, which type of cutting machine did you use? Describe it.
Point 4: Revisions have been made.
Response 4: 2.2 Micromorphology test → 2.2 Microstructure test
Point 5: Revisions have been made.
Response 5: the results were averaged → the mean value was taken, In this way, a total of nine groups of data were obtained. Delete it
Point 6: Revisions have been made.
Response 6: Should be ground to 0.38 μm → should be polished to 0.38 μm
Point 7: Revisions have been made.
Response 7: magnification of 200. → magnification of 200×
Point 8: Revisions have been made.
Response 8: Figure 5. The surface morphology of SEM → Figure 5. The SEM images for the two tested surfaces
Point 9: We deeply apologize for the confusion caused to you by our lack of detail and rigorous presentation. The N in TiN comes from the reaction gas N2 introduced during coating preparation.we have made corresponding changes to the manuscript to make it clearer and smoother. Thank you for your valuable comments, which will be of great help in improving the quality of our manuscript.
Response 9: The elemental 229 mass percentages of Ti and N in the TiN coating were 78.17 and 21.83 respectively. Indicate the source of nitrogen in the coating layer of TiN. This is a very significant point.
Point 10: Revisions have been made.
Response 10: indicating that the main component was Cr.→ indicating that the main component was Cr and the rest is C.
Point 11: Revisions have been made.
Response 11: that the average nano-hard → that the mean value of the nano-hardness
Point 12: Thank you very much for your suggestion. We feel sorry for our poor writing. We have carefully examined the manuscript and revised it. On the basis of this, we have involved native English speakers for language corrections. Revisions have been made, and the title of the full text has been changed.
Response 12: Figure 7. The elastic modulus → Figure 7. The elastic modulus for both case of coating layers.Please reconsider all Figures’ captions for better and clearer description.
Point 13: Your comments are very constructive and we are sorry for the low quality of the Figures. We have replaced all figures with high resolution figures, thank you very much for your valuable comments.
Response 13: Figure 11 . Please make the curves sharper and clearer, and same for Fig. 12, Fig. 15, and Fig 11.
Point 14: Revisions have been made.
Response 14: Figure 13. The surface wear morphology → The SEM micrographs of the wear surfaces with different time.
Point 15: Revisions have been made.
Response 15: TiN spindle was far lower → TiN spindle was much lower
Thank you very much for your suggestion. We feel sorry for our poor writing. Good writing is critical to improve the readability and understandability of the manuscript. We have carefully examined the manuscript and revised it according to English usage. This includes, but is not limited to, replacing repeated or overused words with more specific synonyms to improve the sharpness of the manuscript, fixing grammatical errors, regulating the use of articles and prepositions, removing unnecessary commas, and breaking long sentences that are difficult to understand into two. On the basis of this, we have involved native English speakers for language corrections.
Thank you for your constructive comments. We have reorganized the manuscript after the revisions, which help to improve the overall quality of the manuscript.
According to the comments, we have made revisions to the manuscript. If necessary, we are willing to further improve the manuscript. Thanks for your time and efforts.

Reviewer 2 Report
Dear author,
congratulation for your work.
I still have a few comments:
- please enhance the quality of images from figure 3, 5, 6, 7, 8, 9, 10, 11 and so on (in fact all figures). All of them are of inadequate quality.
- Please add more references to better frame the studied topic
- please describe de used equipment (SEM, friction test equipment, ....)
Author Response
Dear reviewer:
Thank you for your comments concerning our manuscript entitled “Tribological properties of TiN coating on cotton picker spindle” (coatings-2356466). Those comments are valuable and very helpful for revising and improving our manuscript. We have studied the comments carefully and have made the correction. The responses to the comments are as follows:
Point 1: Your comments are very constructive and we are sorry for the low quality of the Figures. We have replaced all figures with high resolution figures, thank you very much for your valuable comments.
Response 1: please enhance the quality of images from figure 3, 5, 6, 7, 8, 9, 10, 11 and so on (in fact all figures). All of them are of inadequate quality.
Point 2: Thank you very much for your constructive comments. We have reformulated the cited work to be more relevant to the research subject of the manuscript.(These valuable works are cited in the introduction section, References [4], [19], [22], [23], [25]).
Response 2: Please add more references to better frame the studied topic
Point 3: We apologize for not providing a description of the instruments used. All instrument names have been added to the manuscript. Thank you for your constructive comments.
Response 3: please describe de used equipment (SEM, friction test equipment, ....)
Thank you for your constructive comments. We have reorganized the manuscript after the revisions, which help to improve the overall quality of the manuscript.
According to the comments, we have made revisions to the manuscript. If necessary, we are willing to further improve the manuscript. Thanks for your time and efforts.

Reviewer 3 Report
The submitted paper deals with an effective application of a PVD TiN coating to a spindle of the horizontal cotton picker for reducing the wear during its operation. Although there are numerous publications in the literature for such coating systems and this work doesn’t overcome the state of the art, the innovation of the submitted paper can be attributed to the coating application. Thus, it is an interesting work and well-written. My remarks are the following:
1) The coating thickness has to be mentioned.
2) Since the coating was deposited on 20CrMnTi steel, the deposition temperature has to be kept low for remaining the substrate properties unaffected. Otherwise, due to annealing the mechanical properties of the substrate worsen. That means this is a low temperature film deposition process. Related comments as well as the deposition temperature have to be mentioned in the manuscript.
3) Since this work may be published in the Journal Coatings, some publications from this Journal in the reference part related to such coating system would be beneficial and give in this work an added-value.
Author Response
Dear reviewer:
Thank you for your comments concerning our manuscript entitled “Tribological properties of TiN coating on cotton picker spindle” (coatings-2356466). Those comments are valuable and very helpful for revising and improving our manuscript. We have studied the comments carefully and have made the correction. The responses to the comments are as follows:
Point 1: Thank you very much for your advice. we have added the coating thickness and we have corresponding modify in the manuscript (Lines 235 to 236). This greatly improved the overall quality of the manuscript.
Response 1: The coating thickness has to be mentioned.
Point 2: Thank you very much for your constructive comments, we apologize for not introducing this point and have supplemented the article accordingly.The following content has been supplemented to the manuscript: The deposition temperature is set to 200 ℃, and a lower deposition temperature has a smaller impact on the matrix performance.
Response 2: Since the coating was deposited on 20CrMnTi steel, the deposition temperature has to be kept low for remaining the substrate properties unaffected. Otherwise, due to annealing the mechanical properties of the substrate worsen. That means this is a low temperature film deposition process. Related comments as well as the deposition temperature have to be mentioned in the manuscript.
Point 3: Thank you very much for your constructive comments. We have reformulated the cited work to be more relevant to the research subject of the manuscript.(These valuable works are cited in the introduction section, References [4], [19], [22], [23], [25]).
Response 3: Since this work may be published in the Journal Coatings, some publications from this Journal in the reference part related to such coating system would be beneficial and give in this work an added-value.
Thank you for your constructive comments. We have reorganized the manuscript after the revisions, which help to improve the overall quality of the manuscript.
According to the comments, we have made revisions to the manuscript. If necessary, we are willing to further improve the manuscript. Thanks for your time and efforts.

Round 2
Reviewer 3 Report
The authors made all the appropriate changes to their manuscript and the submitted paper can be accepted for publication.